

**1**     **Identifying possible Stratification phenomenon in ionospheric F2 Layer**

**2**     **using the data observed by the Demeter satellite: Method and Results**

**3**     Xiuying Wang, Dehe Yang, Dapeng Liu, Wei Chu

**4**     Institute of Crustal Dynamics, China Earthquake Administration, Beijing, China

**5**     Corresponding author:   Xiuying Wang (652383915@qq.com)

**7**     Abstract: Many studies have revealed the stratification phenomenon of the topside ionospheric

**8**     F2 layer using ground-based or satellite-based ionograms, which can show direct signs of this

**9**     phenomenon. However, it is difficult to identify this phenomenon using the satellite-based in situ

**10**     electron density data. Therefore, a statistical method, using the shuffle resampling skill, is

**11**     adopted in this paper. For the first time, in situ electron density data, recorded by the same

**12**     Langmuir probe onboard the Demeter satellite at different altitudes, are analyzed and a possible

**13**     stratification phenomenon is identified using the proposed method. Our results show that the

**14**     nighttime stratification, possibly a permanent phenomenon, can cover most longitudes near the

**15**     geomagnetic equator, which is not found from the daytime data. The arch-like nighttime

**16**     stratification decreases slowly on the summer hemisphere and thus extends a larger latitudinal

**17**     distance from the geomagnetic equator. All results, obtained by the proposed method, indicate

**18**     that the stratification phenomenon is more complex than what has previously been found. The

**19**     proposed method thus is an effective one, which can also be used on similar studies of

**20**     comparing fluctuated data.

**21**

**22**     Key words: stratification, ionospheric F2 layer, in situ electron density, Demeter satellite,

**23**     significance test

**24**     **1 Introduction**

**25**     Stratification of the F2 layer, an enhancement in electron density at heights above the F2 layer

**26**     maximum in the ionosphere at low latitudes and mid-latitudes, was first reported in the mid-

**27**     twentieth century (Heisler, 1962; Sen, 1949; Skinner et al., 1954). Sayers et al. (1963) was then the

**28**     first to detect topside ledges in the equatorial ionosphere using a Langmuir probe onboard the

**29**     Ariel-I satellite and predicted that the topside ionograms would reveal the ledges as cusps, as later

**30**     proved by many studies using the topside sounding technique (Lockwood & Nelms, 1964;

**31**     Raghavarao & Sivaraman, 1974; Sharma & Raghavarao, 1989).

**32**     There were few studies of the stratification phenomenon until the mid-1990s. Balan and Bailey

**33**     (1995) then explained the formation mechanism of the F3 layer using the SUPIM (Sheffield

**34**     University Plasmasphere–Ionosphere Model). They referred to the layer as G layer and was later

**35**     renamed as F3 layer because it has the same chemical composition as the F region (Balan et al.,

**36**     1997). Since then, many more studies on the mechanism and spatial and temporal distributions of

**37**     the phenomenon have been carried out (Batista et al., 2002; Depuev & Jenkins, 1997; Depuev &

**38**     Pulinets, 2001; Hsiao et al., 2001; Rama Rao et al., 2005; Tardelli et al., 2016; Uemoto et al., 2007;

**39**     Zain et al., 2008; Zhao et al., 2011a, 2001b).

**40**     However, most research has used ionograms or total electron content data recorded on the

**41**     ground (Balan et al., 1998; Bastica et al. 2002; Jenkins et al., 1997; Nayak et al., 2014; Rama Rao et



al., 2005; Zhao et al., 2011a), where the distribution features of the stratification phenomenon
cannot be obtained because only data of discontinuously distributed observation stations can be
used. Studies on the stratification of the F2 layer at the topside ionosphere were therefore carried
out using sounding techniques onboard low-Earth-orbiting satellites (Karpachev et al., 2012;
Thampi et al., 2005; Uemoto et al., 2004, 2006; Zhao et al., 2011b). Topside ionograms can reveal
the occurrence of the F3 layer when the peak electron density of the F3 layer, namely $N_mF_3$, is
smaller than $N_mF_2$, which cannot be observed using an ionosonde on the ground. However, the
short-term global scale distribution of the stratification phenomenon still cannot be obtained from
satellite-based ionograms even though such ionograms can provide more data because the
obtained data are still discontinuous.

In addition, nearly all the above-mentioned F2 layer stratification studies were carried out using
indirect observation data, in which case some detailed information may be missed. A method
therefore is proposed in this paper, which can compare the in situ electron density data obtained
at different altitudes and identify their differences. Based on this method, the in situ electron
density data, recorded by the Demeter satellite at the topside ionosphere, is used to study the
stratification phenomenon, enabling us to investigate the characteristics of the global-scale
distribution and other information about the stratification phenomenon.

The result that the electron density observed at higher altitude is greater than that observed
at lower altitude suggests a stratification phenomenon distributed in a large area. This result was
obtained using in situ electron density data obtained before and after an altitude adjustment of
the Demeter satellite in a relatively short time, which is the first direct comparison of in situ data
recorded by the same instrument but at different altitudes. The results of the distribution features
of this phenomenon, obtained by the proposed method, are in accord with those obtained by
previous studies, but some features also suggest that the stratification phenomenon is more
complicated than previously found, thus demonstrating that the proposed method is effective.

## 2 Data and Method

### 2.1 Data

The data used in this study were obtained from Demeter (Detection of Electro-Magnetic
Emission Transmitted from Earthquake Regions), a French micro-satellite operated by CNES (Centre
National d'Etudes Spatiales) and devoted to the investigation of ionospheric disturbances due to
seismic, volcanic and tsunami activities. The Demeter satellite was launched in June 2004.
Observation data were recorded from the end of November 2004 to December 2010. Owing to its
specific orbit, Demeter is always located at about 10:30 or 22:30 local time. The satellite made
continuous measurements between invariant latitudes of −65° and +65°. The ISL (Instrument
Sonde de Langmuir) is one of the five scientific payloads and recorded in situ data of the electron
density, ion density and electron temperature (Lagoutte et al., 2006; Lebreton et al., 2006).

The Demeter satellite adjusted its flying altitude in its initial flight stage and between the end
of 2005 and the beginning of 2006, as shown in Fig. 1, which presents the average flight altitude of
the ascending (nighttime) and descending (daytime) orbit between southern and northern
geographical latitudes of 50° from November 17, 2004 to December 31, 2006.

The history of the altitude of the satellite can be divided into four stages.
(1)  The altitude of the satellite was not fixed but varied between about 703 and 725 km from



November 17, 2004 to March 10, 2005.
(2)  The average orbital altitude was fixed at around 709 km after March 10, 2005.
(3)  The average altitude was adjusted to approximately 677 km from January 1 to 9, 2006.
(4)  The altitude was fixed at an average value of about 669 km from January 14, 2006.

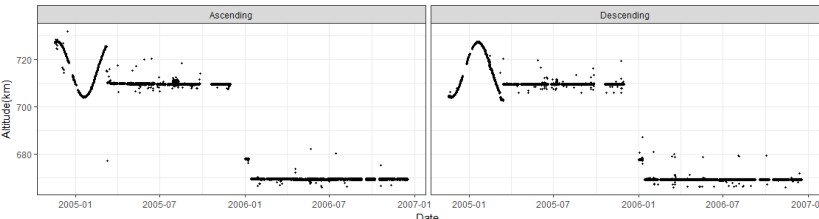


Fig. 1 Average altitude of the Demeter satellite from November 2004 to December 2006
The data recorded by the Demeter satellite before and after its altitude adjustment provide
an opportunity to study the vertical gradients of electron density in a small height range of the
topside ionosphere using in situ electron density data recorded by the same instrument. Since the
altitude of the satellite was not fixed at a constant value from November 2004 to March 2005, and
there was no data in December 2005, data recorded before and after the adjustment at the
beginning of 2006 are selected in the study; during this periods, the orbit altitude was respectively
fixed at 677 and 669 km.
The geomagnetic index Dst and the solar activity index F10.7 in January 2006 are presented
in Fig. 2. The figure shows geomagnetically quiet days from January 1 to 25, 2006, and the F10.7
index of solar activity before altitude adjustment was roughly equal to or smaller than that after
the adjustment. Therefore, data from January 1 to 25, 2006 will be used in this paper, because the
differences in geomagnetic and solar influences are negligible during this period.

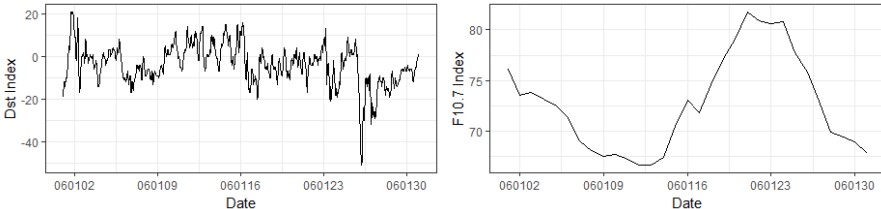


Fig. 2 Geomagnetic index Dst and solar activity index F10.7 in January 2006
Many studies have shown that the electron density in the F2 layer is characterized by periodic
changes in the diurnal, seasonal, annual and solar activity cycles and fluctuations due to other
random factors, such as geomagnetic storms and sunspot eruptions. Issues therefore need to be
addressed before carrying out this study.
As mentioned above, the local time that the Demeter Satellite passed over a location was
roughly fixed at about 10:30 in the morning and about 22:30 in the evening, which means that
diurnal changes in the data can be ignored when comparing the data before and after the altitude
adjustment at the same place because the local time is consistent. Another issue, which is the focus
of this study, is that when the electron density data are recorded over a relatively short time under
quiet observation conditions, say a few days, variations due to the long-period trend in the data
(e.g., seasonal and annual variations) can be ignored, that is to say, the data observed in a few days
is usually similar to that observed a few days ago.



Against this background, the data, observed before and after the altitude adjustment of the
Demeter satellite in a relatively very short time, are compared and analyzed by seeking a suitable
mathematical method.
## 2.2 Method
The electron density is known to dynamically change both spatially and temporally. It is
therefore uncertain that the difference before and after the adjustment of the orbital altitude is
the result of normal data fluctuation or the result of the altitude adjustment. It is necessary to
design a reasonable scheme with which to distinguish the cause of the difference.
A significance test is a statistical method of determining whether the difference between two
groups of data is significant. Employing this method, if the $p$-value, the probability that a given
result occurs under the null hypothesis of no difference between the two groups, is less than a
predefined significance level, then the null hypothesis is rejected at the chosen level of significance
and the alternative hypothesis of a difference between the two groups is accepted. However, if the
$p$-value is not less than the chosen significance threshold, then the evidence is insufficient to
support a conclusion. Significance tests can therefore be conducted to determine whether the
difference between before and after the adjustment of the altitude can be ascribed to the
randomness of the data variation. If not, it may be caused by the altitude adjustment because all
the other conditions are the same.
However, the significance test assumes data to be normally distributed, which the electron
density data are not. This paper thus conducts a permutation test (Hesterberg et al., 2003), a
distribution-independent computer simulation approach of resampling advised by Fisher and Yates
(Wikipedia).
The basic idea of the permutation test is to resample the data many times to check whether
the same pattern of results is observed if the observation data are randomly assigned to
experimental groups. If the statistics calculated from the obtained data fall outside the confidence
limits, say 95%, the observed difference is far out in the left or right tail, and one can conclude that
there is a significant difference between the groups. A permutation test is based on available data
rather than a set of standard assumptions about underlying populations. It is therefore distinct
from traditional statistics and can give accurate $p$-values with which to check the significance of
the difference between two data groups.
We therefore adopt the permutation test method to compare the data observed at different
altitudes by the Demeter satellite, and to check whether the differences between the data
observed at different altitudes are significant. Using this method, the general process of data
analysis in this study is as follows.
(1)  Construct data groups using the data observed before and after the altitude adjustment,
or data observed at same altitude.

➢  Divide the area covered by the satellite orbit between latitudes of 50° south and
50° north into cells of 5° latitude and 10° longitude.

➢  Calculate the mean electron density before and after the adjustment of altitude in
each cell.

➢  Divide the data into different regions every 5° latitude and obtain 20 regions from
50° south to 50° north in the latitudinal direction.

(2)  Compare the data groups constructed from observation at different altitudes and check



the significance of their differences by employing the permutation test method.
(3) Compare the data groups constructed from observation at similar conditions but with
same altitude and check the significance of their differences as a reference.
(4) Draw conclusions by analyzing different results.
A uniform significance level of 0.05 and one-side test are adopted in this paper, and no special
explanation is given in the following.

**3 Data comparison**

3.1 Data construction

According to Section 2.1, the data obtained from January 1 to 25, 2016 is selected to carry out
the analysis. During this period, the data from January 1 to 9 was obtained before the altitude
adjustment, and the data from January 14 to 25 was obtained after the altitude adjustment. In
addition, the geomagnetic and solar activity indices were every low during this period; that is, the
data obtained before and after the altitude adjustment were measured under similar observation
conditions.
In order to construct the data groups for comparison, a scheme is designed to divide the data
into different groups. Ascending data (data recorded during the night) from January 1 to 8 and from
January 15 to 23, 2006, are both divided into two groups, to give a total of four groups of data with
each having equal observation days. Details of the grouping are given in Table 1.
Table 1 Grouping information of the data from January 1 to 23, 2006

| Group No. | Date of observation | Average Altitude | Altitude Adjustment |
|---|---|---|---|
| Group 1 | 1, 2, 3, 4 | 677.76km | Before |
| Group 2 | 5, 6, 7, 8 | 677.78km | Before |
| Group 3 | 15, 16, 17, 18 | 669.34km | After |
| Group 4 | 20, 21, 22, 23 | 669.33km | After |

Based on this grouping scheme, comparative data are constructed using the cells of 5° in the
latitudinal direction and 10° in the longitudinal direction as mentioned in section 2.2. The average
value of the recorded data in each cell is computed using data from Group 1 to Group 4; there are
thus 36 cells × 4 groups of data for each latitudinal region. Data analysis involves comparing the
data between groups in each latitudinal region, including both the cases of data comparisons
between different altitudes and between the same altitudes.

3.2 Comparison in one latitudinal region

The four groups of data, in the region of geographical latitude −5° to 0°, are compared with
each other as a demonstrative example of the proposed method.
In order to determine the differences between two groups of data are caused by random data
fluctuation or by altitude differences, significance tests are carried out for each pair of groups using
the improved Fisher–Yates permutation test method (Durstenfeld, 1964), in which the distribution
of the mean data difference is obtained by resampling the data 10,000 times. The actual mean data
differences of each pair of groups are then compared with the 5% confidence level of the
corresponding distribution.
The significance test results of each pair of groups using the data located in geographical
latitude (−5, 0) are shown in Fig. 3, and the corresponding permutation test $p$-values are given in



Table 2.

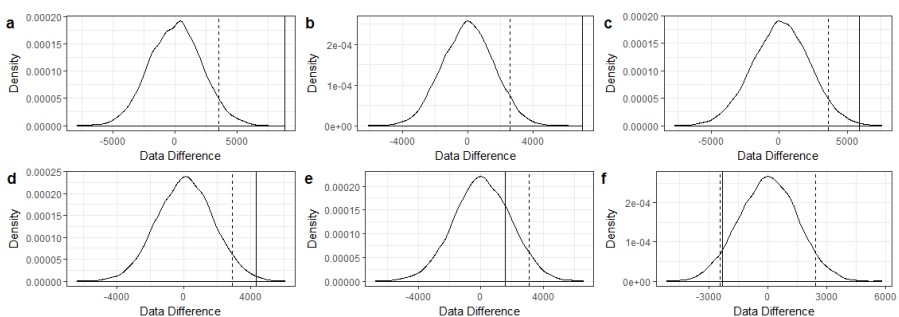


Fig. 3 Density distributions of the mean difference obtained in the permutation test

(The dashed line is the mean difference corresponding to the 5% confidence level while the solid line is the
observed mean difference between two groups. The lower 5% confidence level is also shown for f because the
data difference is negative. Here, a is the permutation test result for Groups 1 and 3; b, Groups 2 and 3; c, Groups
1 and 4; d, Groups 2 and 4; e, Groups 1 and 2; f, Groups 3 and 4.)

Table 2 Permutation test results of ascending data at a geographical latitude of −5° to 0°

| Latitude region | Group1-3 | | Group 2-3 | | Group 1-4 | | Group 2-4 | | Group 1-2 | | Group 3-4 | |
|---|---|---|---|---|---|---|---|---|---|---|---|---|
| | $M_{Diff}$ | p | $M_{Diff}$ | p | $M_{Diff}$ | p | $M_{Diff}$ | p | $M_{Diff}$ | p | $M_{Diff}$ | p |
| (-5,0) | 8797.95 | 0.0000 | 7031.11 | 0.0000 | 5909.50 | 0.0025 | 4325.02 | 0.0049 | 1584.48 | 0.2136 | -2312.43 | 0.0593 |

($M_{Diff}$ represents the mean of differences between two groups, while *p* is the probability that the mean data
difference calculated in the permutation simulation is greater than the observed $M_{Diff}$ if it is positive or less than
the observed $M_{Diff}$ if it is negative.)

In Fig. 3, the solid lines represent mean values of data differences before and after the altitude

adjustment in each cell:

$$M_{Diff} = \frac{1}{N}\sum_{i=1}^{N}(B_i - A_i) = \frac{1}{N}\sum_{i=1}^{N}B_i - \frac{1}{N}\sum_{i=1}^{N}A_i. \qquad (1)$$

Here, N is total number of cells in each latitude region, B is the average value in cell i before

altitude adjustment, and A is the average value in the same cell after the adjustment. Equation (1)
shows that the mean value of data differences is equal to the data difference between average
values of all cells before and after the adjustment. Therefore, mean values of data differences can
be calculated using two average values. As shown in Fig. 3, the data differences, between the
average data in the two groups in random permutation tests conducted 10,000 times, follow a
normal distribution with a mean value of zero, and the probability of the occurrence of the original
data difference is zero or extremely small, which indicates that data recorded before the
adjustment in most cells are obviously greater than those recorded after the adjustment because
the mean differences are much greater than zero.

Figure 3 and Table 2 show that the differences between Groups 1 and 3, Groups 2 and 3,

Groups 1 and 4, and Groups 2 and 4, representing the differences before and after the adjustment
of altitude, are significant because the *p*-values are zero or close to zero, much less than the
predefined significance level of 5%. This means that the likelihood of observing the actual data
difference given that the two groups have no difference is unlikely. Therefore, the null hypothesis
of no difference can be rejected, and significant difference between the two groups is determined.
Meanwhile, the *p*-values of Groups 1 and 2 and Groups 3 and 4, representing differences at the

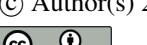



same altitude before and after the adjustment respectively, are greater than the predefined
significance level, which means the difference between the two groups is not significant and the
hypothesis of no difference between the two groups cannot be rejected.

The permutation test results of data at different altitudes and data at similar altitudes show a
significant contrast, indicating that the significant differences between the data before and after
the adjustment are by no means accidental but due to potential causes. Moreover, an interesting
point is that the electron density data recorded at higher altitude is higher than that of lower
altitude because all differences (i.e., values before adjustment minus values after adjustment) are
positive, different from the normal attenuation law at the topside ionosphere, which implies the
possible stratification phenomenon during the selected time segment.

### 3.2 Comparison in all latitudinal region

Obvious difference between the data groups in one latitudinal region show some information.
To obtain the distribution of this significant difference, permutation test results for the 20 regions
from 50° south to 50° north in geographical and geomagnetic latitude (where the geomagnetic
latitude refers to the dipole coordinates given in the Demeter satellite dataset) are obtained, and
the variations of $p$-values with latitude are presented in Fig. 4. Table 3 only gives the permutation
test results in geomagnetic latitudes because the results calculated from geographical latitudes are
similar to those of geomagnetic latitudes.

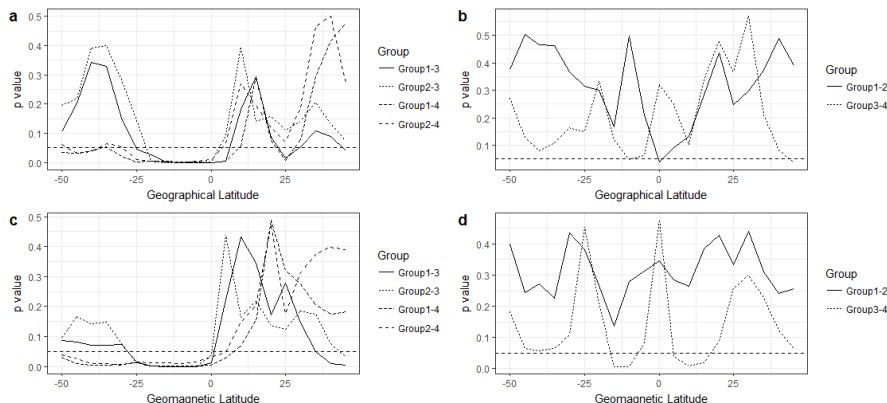


Fig. 4 Variations of $p$-values with geographical/ geomagnetic latitude

Table 3 Permutation test results of ascending data in the 20 geomagnetic latitude regions

| Latitude region | Group1-3 | | Group 2-3 | | Group 1-4 | | Group 2-4 | | Group 1-2 | | Group 3-4 | |
|---|---|---|---|---|---|---|---|---|---|---|---|---|
| | $M_{Diff}$ | p | $M_{Diff}$ | p | $M_{Diff}$ | p | $M_{Diff}$ | p | $M_{Diff}$ | p | $M_{Diff}$ | p |
| 45,50 | -2258.44 | 0.0040 | -1717.80 | 0.0299 | -839.74 | 0.1835 | -299.10 | 0.3889 | -540.64 | 0.2569 | 1418.71 | 0.0639 |
| 40,45 | -1950.41 | 0.0091 | -1335.83 | 0.0770 | -903.82 | 0.1739 | -289.24 | 0.3966 | -614.58 | 0.2418 | 1046.59 | 0.1243 |
| 35,40 | -1184.69 | 0.0507 | -810.07 | 0.1748 | -718.42 | 0.2075 | -343.80 | 0.3748 | -374.62 | 0.3093 | 466.27 | 0.2284 |
| 30,35 | -868.94 | 0.1464 | -770.15 | 0.1845 | -473.54 | 0.2745 | -378.75 | 0.3120 | -98.79 | 0.4401 | 279.48 | 0.2990 |
| 25,30 | -578.48 | 0.2779 | -822.49 | 0.1246 | -372.14 | 0.3199 | -585.42 | 0.1744 | 213.29 | 0.3327 | 324.59 | 0.2562 |
| 20,25 | -901.29 | 0.1727 | -975.20 | 0.1385 | -28.67 | 0.4882 | -72.08 | 0.4747 | 43.41 | 0.4284 | 966.15 | 0.0866 |
| 15,20 | -600.00 | 0.3450 | -986.37 | 0.2207 | 1347.41 | 0.1533 | 961.04 | 0.2034 | 386.37 | 0.3848 | 1947.41 | 0.0185 |
| 10,15 | -269.15 | 0.4330 | -1374.58 | 0.1644 | 2329.54 | 0.0676 | 1224.11 | 0.1490 | 1105.43 | 0.2632 | 2598.68 | 0.0082 |
| 5,10 | 1374.33 | 0.2236 | 237.71 | 0.4381 | 3227.69 | 0.0272 | 2283.55 | 0.0466 | 1136.61 | 0.2854 | 2253.04 | 0.0395 |
| 0,5 | 4013.46 | 0.0112 | 3112.33 | 0.0373 | 4305.87 | 0.0052 | 3404.74 | 0.0302 | 865.87 | 0.3455 | 292.40 | 0.4765 |
| -5,0 | 6854.30 | 0.0000 | 5875.57 | 0.0002 | 4616.65 | 0.0024 | 3791.28 | 0.0150 | 825.37 | 0.3137 | -1747.23 | 0.0792 |
| -10,-5 | 8723.66 | 0.0000 | 7919.08 | 0.0000 | 4863.00 | 0.0013 | 4219.73 | 0.0107 | 643.27 | 0.2788 | -3586.31 | 0.0069 |

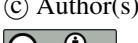



| -15,-10 | 9649.68 | 0.0000 | 7727.16 | 0.0000 | 5994.04 | 0.0013 | 4071.51 | 0.0129 | 1922.53 | 0.1363 | -3655.64 | 0.0069 |
| -20,-15 | 7437.61 | 0.0003 | 6051.83 | 0.0011 | 6151.21 | 0.0017 | 4279.33 | 0.0119 | 1385.78 | 0.2656 | -1481.91 | 0.2102 |
| -25,-20 | 4618.32 | 0.0148 | 4044.56 | 0.0118 | 4679.33 | 0.0118 | 3894.36 | 0.0152 | 573.76 | 0.3826 | 80.28 | 0.4544 |
| -30,-25 | 2682.88 | 0.0741 | 2609.11 | 0.0766 | 4594.21 | 0.0060 | 4408.26 | 0.0056 | 185.95 | 0.4363 | 1884.87 | 0.1072 |
| -35,-30 | 2792.20 | 0.0717 | 1732.04 | 0.1484 | 5047.46 | 0.0034 | 3864.55 | 0.0083 | 1182.90 | 0.2264 | 2257.67 | 0.0655 |
| -40,-35 | 2560.95 | 0.0711 | 1597.18 | 0.1422 | 4972.26 | 0.0040 | 4008.49 | 0.0097 | 963.76 | 0.2713 | 2411.31 | 0.0564 |
| -45,-40 | 2449.71 | 0.0804 | 1420.34 | 0.1663 | 5032.67 | 0.0086 | 3779.51 | 0.0258 | 1198.12 | 0.2432 | 2573.52 | 0.0640 |
| -50,-45 | 2701.66 | 0.0879 | 2697.81 | 0.0934 | 4126.46 | 0.0300 | 4025.30 | 0.0377 | 94.91 | 0.4008 | 1601.15 | 0.1813 |

The permutation test results in Figs. 4 and Table 3 have obvious regular distribution patterns.

(1) There are significant differences in data only before and after the adjustment of altitude in continuous latitudinal regions; i.e., there are significant differences in data between Groups 1 and 3, Groups 2 and 3, Groups 1 and 4, and Groups 2 and 4. Meanwhile, the differences between observation data for the same orbital altitude, namely differences between Groups 1 and 2 and Groups 3 and 4, are not obvious and no regular distribution pattern exists in the data.

(2) The data having a statistically significant difference are mainly distributed near the geographical or geomagnetic equator regions, and are more skewed towards the Southern Hemisphere, where the time of the observation data is just summer.

(3) Comparing the distribution of data with significant differences in Figs. 4, it is seen that the distribution is 5° south in geomagnetic latitude, which indicates that this regular distribution of the data with significant differences may be mainly controlled by the geomagnetic latitude, and the regular distribution in terms of the geographical latitude is due to the distribution region in geographical latitude overlapping with regions beside the geomagnetic equator.

(4) Table 3 shows that the data differences change from being positive from lower to higher mid-latitudes in the Southern Hemisphere to being negative in the corresponding latitudes in the Northern Hemisphere, just like an arch extending toward the higher latitudinal direction in both hemispheres, as shown in Fig. 5. This regular distribution cannot be a coincidence, because although most $p$-values in the mid-latitude regions do not reject the null hypothesis of no significant difference between the data observed at different altitudes, the probability that positive differences appear simultaneously in several continuously latitudinal regions (multiplication of the $p$ values in each latitudinal region) is extremely low according to the obtained $p$-values, which indicates an underlying control factor. Regarding all differences in the Northern (winter) Hemisphere being negative, this is the normal attenuation pattern of the F2 layer.

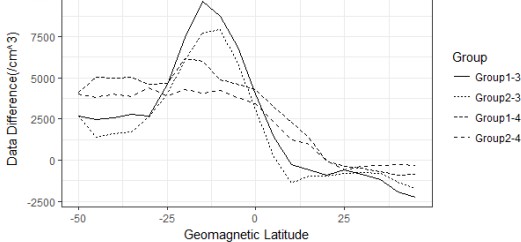

Fig. 5 Variations of data differences with geomagnetic latitude

The distribution characteristic, that data with significant differences are distributed in the vicinity of the geomagnetic equator, is consistent with the regions where stratification of the F2 layer has been found in many studies, and the stratification phenomenon can exactly explain the electron density at higher altitude being greater than that at lower altitude.





Figure 6 presents all the regular patterns summarized above using the average electron
density data of the four groups before and after altitude adjustment in each latitudinal region. The
figure shows that the curves of the average electron density data vary with latitude, with the
maximum differences being located at about 10 in the southern hemisphere.

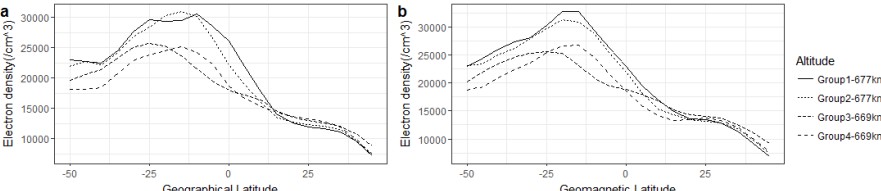

Fig. 6 Variation of the average ascending electron density with latitude

Figure 6 shows that the difference between the two groups of data before the adjustment of
the orbital altitude, namely Groups 1 and 2, is small while the difference between the two groups
after the adjustment, namely Groups 3 and 4, is also small. However, when comparing the four
groups together, obvious differences between the groups before and after the adjustment are seen
in the vicinity of a geographical latitude of −10° or a geomagnetic latitude of −15°. Moreover,
the difference is more pronounced in the Southern Hemisphere than in the Northern Hemisphere.
Although the greater data fluctuations in the summer Southern Hemisphere are a cause of this
phenomenon, the regular distribution cannot be explained by random fluctuation in the data.

### 3.4 Reference Comparisons

To further demonstrate that the phenomenon found above is caused by non-random factors,
several sets of data other than the above mentioned data are constructed to compare whether the
same regular distribution patterns can be found.
**1. Descending data for the same period**
The permutation test results of descending data, data recorded during the day, are calculated
according to the grouping information in Table 1. The results show that there are both cases of
significant differences and insignificant differences between the data observed at different
altitudes and between the data observed at same altitudes. Variations in the average electron
density with latitude are given in Fig. 7. The figure clearly shows that the observation data for the
same altitude during the day fluctuate greatly and there are no consistent regularities among
different data groups. Therefore, although there are cases that a higher altitude has higher electron
density, a definite conclusion cannot be drawn from these descending data.

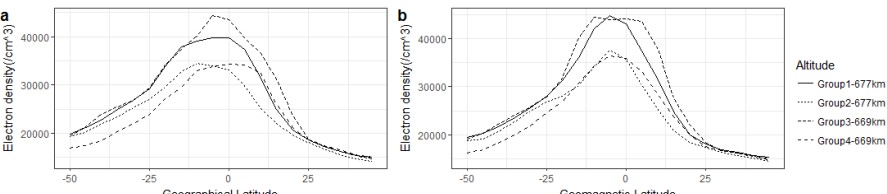

Fig. 7 Variation of the average descending electron density with latitude

**2. Ascending data in different periods**
Besides the above analysis, groups of reference data are also calculated to further confirm



that the regular distributions in Fig. 4 are not accidental. Because there were small geomagnetic
storms in January 2007 and 2008, only data in 2009 and 2010 are used here for comparison. Data
groups, with the same geomagnetic and grouping conditions and using the ascending data (data
observed during nighttime) for 2009 and 2010, are calculated using the permutation test method.
Figures 8 and 9 show the variations of ascending electron density data with geographical/
geomagnetic latitude using the data recorded in 2009 and 2010 respectively; no obvious
differences are found from these data. Therefore, the significant differences shown in Figs. 4 are
not coincidental.

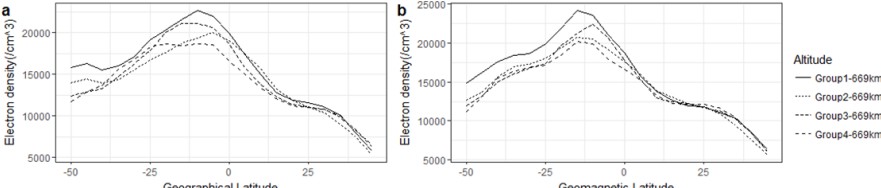


Fig. 8 Variation in the ascending electron density with latitude obtained using data for 2009

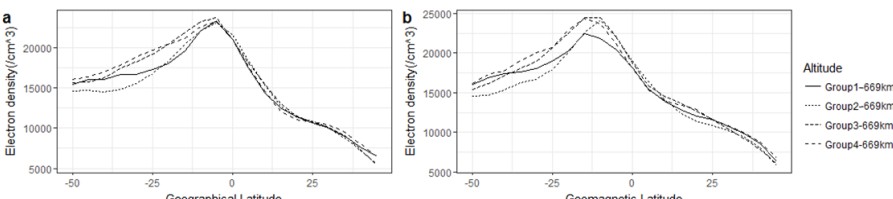


Fig. 9 Variation of the ascending electron density with latitude obtained using data for 2010

## 4 Discussion

We conclude from the above data analysis that the phenomenon that the in situ electron
density observed at higher altitude is greater than that observed at lower altitude and that
significant differences are distributed regularly in the vicinity of the geomagnetic equator on a
global scale, is the stratification phenomenon of the F2 layer. Although the data were not
recorded at the same time, the data variation can be neglected because the time interval is short
and observing conditions are similar.
According to the data grouping and calculation method, if the phenomenon is only due to
random data fluctuation, the possibility that this phenomenon appears only for data recorded at
different altitudes and at several latitudinal regions in the vicinity of the geomagnetic equator at
the same time is extremely low. Moreover, the same regular distribution from data recorded at
other times with similar grouping conditions cannot be observed. The possibility that the regular
data distribution is due to random factors can therefore be excluded definitely.
In addition, the significant difference between two data groups before and after the altitude
adjustment near the geomagnetic equator region indicates that most data in the 36 cells in each
latitudinal region have a significant difference. It is thus deduced from the data that the
stratification phenomenon in the F2 layer covers a large longitudinal area near the geomagnetic
equator region. This is different from the conclusion of those studies (Balan et al., 1998; Rama
Rao et al., 2005; Zhao et al., 2011a) that the phenomenon can only be observed at special
longitudes, which may be due to the fact that the peak of the stratification is less than that of the



F2 layer in most of the longitudinal area for most of the time, and thus invisible to the ground-
based observation.
In fact, the stratification phenomenon has been observed at many locations using ionosonde;
e.g., Brazil (Balan et al., 1997, 1998, 2000; Batisca et al., 2002; Jenkins et al., 1997), Southeast
Asia (Hsiao et al., 2001; Lynn et al., 2000), India (Rama Rao et al., 2005; Thampi et al., 2005) and
China (Jiang et al., 2015), illustrating that the stratification phenomenon is distributed across a
large longitudinal area in spite of the scatter discoveries. The study of Zhao et al.( 2011b) using
long-time satellite-based ionograms also showed that the stratification is distributed in all
longitudinal areas along the magnetic equator. The results obtained from the in situ data are
thus in accordance with the results of those studies, and further approve that this phenomenon
may be continuous distributed along the longitudinal direction. The global scale in situ electron
density data of the Demeter satellite observed in a short time provides an opportunity to study
the distribution features of the stratification phenomenon, which are difficult to detect through
scattered ground-based or satellite-based sounding data.
Section 3 showed that the recording time of the data used in this study, namely the time of the
stratification, happened to coincide with the downward cycle of the 23rd solar cycle, when the
solar activity was relatively low. The season of stratification found in the data in this study
coincided with summer in the Southern Hemisphere, and the stratification was almost entirely
located in the Southern Hemisphere in terms of the geomagnetic latitude. These spatial and
temporal distribution characteristics, distinct on the summer side of low solar activity, are exactly
the same as those of the F2 layer stratification phenomenon obtained in many studies (Balan et
al., 1998; Batista et al., 2002; Nayak et al., 2014; Rama Rao et al., 2005; Sharma & Raghavarao,
1989)

As for the local time at which stratification occurs, many studies have suggested that the
stratification phenomenon mainly occurs during the day, just as Balan et al. (1998) reported that
the F3 layer occurs mainly during the morning–noon period owing to the combined effect of the
upward E × B drift and neutral wind that provides upward plasma drifts at and above the F2
layer. However, more and more studies have confirmed the existence of nighttime stratification.
Zhao et al. (2011a) studied the post-sunset stratification phenomenon and suggested that the
sunset F3 layer should be distinguished from the traditional morning–noon F3 layer. Lockwood
and Nelms (1964) suggested that the stratification of the F layer can be observed until about
local midnight using the topside sounder data of the ionogram onboard the Alouette satellite.
Karpachev et al. (2012) examined the large data set of IK-19 and found that the F3 layer can
permanently exist until 02:00–03:00 LT. Nevertheless, the F3 layer is rarely recorded at night.
Depuev and Pulinets (2001) also found midnight stratification and showed that the critical
frequency of the nocturnal F3 layer is always essentially lower than $f_0F_2$. It is thus impossible to
observe midnight stratification from the bottom side. They also reported that the real peak
height ($h_mF_3$) of the F3 layer defined by electron density profiles varied from 670 to 730 km.
Rama Rao et al. (2005) pointed out that the altitude of the F3 layer is high at the magnetic
equator (600–700 km). The altitude of the stratification in these studies is almost the same as
the altitudes of the in situ data used in this paper.
Klimenko et al. (2012) suggested that the formation mechanism of additional layers in the
equatorial ionosphere is due to the action of the non-uniform in height zonal electric field at the
geomagnetic equator, and can happen at any time, which can explain the occurrences of the F3



layer and multilayer at different local times, especially at night.

An interesting point, which has not been discussed in earlier studies, is that all differences in

each latitude region on the summer hemisphere are positive though some do not pass a
significance test. This consistent distribution cannot be obtained if data fluctuate randomly. We
therefore speculate that this feature may be related with the stratification phenomenon and
small stratification may exist in the summer hemisphere a little distance away from the
traditional geomagnetic equator region of stratification.

Summarizing the above discussions, we believe that the results obtained in this paper are the

stratification phenomenon in the ionospheric F2 layer, and the proposed method is effective.
The results of this method indicate that the stratification phenomenon may extend to a larger
area in the summer hemisphere, but it is difficult to detect because the differences are small.
The distribution features obtained by the data analytic results also indicate that the stratification
phenomenon is more complex than what has been found previously.

## 398    5 Conclusion

To compare the in situ electron density data observed by the Demeter Satellite at different

altitudes, a statistical method, using the permutation resampling skill, is adopted and used to carry
out the data comparison and analysis work. The results of 10,000 permutation tests, using the
ascending data (data observed during nighttime) obtained before and after the altitude
adjustment, show that there are significant differences between data recorded at different
altitudes near the geomagnetic equator, but no significant differences can be found from the
multiple reference datasets. The stratification phenomenon can explain the regular distribution
patterns summarized from the data analytic results. In addition, the location, altitude, season and
local time of this phenomenon are accordance with the results of many studies on the F2 layer
stratification phenomenon. We therefore believe that the significant difference between the
observations of the Demeter satellite at different altitudes is the stratification phenomenon, and
the proposed method is effective and applicable to similar data analytic studies.

Some features of the stratification phenomenon can also be summarized from the data

analysis results.
1.   The possible stratification phenomenon is found from the nighttime data but cannot be

obtained from the corresponding daytime data, though many studies have pointed out

that this phenomenon occurs mainly during the day, which implies the nighttime

stratification may be a permanent phenomenon.

2.   The phenomenon can occur in most longitudinal regions, which is not in accordance with

the finding of studies that the phenomenon can only appear in special longitudinal

regions. This may be due to the peak of the stratification being less than $f_0F_2$ in most

longitudinal regions for most of the time.

3.   The significance of differences decreases with latitude away from the geomagnetic

equator, indicating that the stratification is just as an arch along the latitude.

4.   Data differences, all of which are positive at lower to higher mid-latitudes in the summer

hemisphere, indicate that the latitudinal extent of the stratification phenomenon is much

larger in the summer hemisphere than the winter hemisphere and small stratification

may exist away from the traditional stratification region. Stratification phenomenon is

more complex than what has previously been found.



## Acknowledgment

This work is supported by National Key R&D Program of China under Grant no. 2018YFC1503505, by Beijing Natural Science Foundation under grants no. 8184091, and by the foundation of Institute of Crustal Dynamics, CEA under grants no. ZDJ2017-20. The electron density data used in this paper are available from the Demeter Data Server (demeter.cnrs-orleans.fr).

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
