# Peer review of "Identifying possible Stratification phenomenon in ionospheric F2 Layer using the data observed by the DEMETER satellite: Method and Results"

_Annales Geophysicae, 2019_

## Referee Comment (RC1) · Angelo De Santis (Referee) · 8 May 2019

This paper concerns the study of the electron density from DEMETER satellite (2004-2010) during the change of altitude (January 2006). The way to establish a possible F2 stratification is quite original, although it cannot provide a definitive response to the question (probably this should be better evidenced in the text).

Therefore, the paper deserves publication, with only a very minor corrections, as follows:

- Please write, in the title and across the whole text, DEMETER with capital letters,

[Figure]

being an acronym.

- Please remove "random" at line 106: the phenomena as geomagnetic storms and sunspot eruptions are not completely random, having some components of quasi-periodicity (27 days, 11 years, etc.).

- Please correct at line 167: "2016" with "2006".

---

## Referee Comment (RC2) · Galina Gordienko (Referee) · 20 May 2019

In this work the authors investigate the stratification phenomenon in the ionospheric F2 layer using the nighttime data observed by Demeter satellite in the period from January 1 to 25, 2006. The data were recorded by the Demeter satellite before and after its altitude adjustment that provide to the authors an opportunity to study the vertical gradients of electron density in a small height range (between ïA¿677.8 and ïA¿669.3 km) of the topside ionosphere using in situ electron density data recorded by the same instrument. It is very important that the period of quiet days (Dst = ±20 nT; F10.7 = 70 ± 5) was chosen by the authors. An original statistical method proposed by authors is

used to study the stratification phenomenon. The data observed at different altitudes by the Demeter satellite are compared, and the significance of the differences is checked. As a result, it was found that the electron density data recorded at higher altitude are higher than those of lower altitude, a feature of the stratification phenomenon. Finally, it was found that the stratification occurs in the vicinity of the geomagnetic equator on a global scale that was interpreted by authors as the stratification of the F2 layer. The paper presents interesting original results and valuable observations which should be published in the "Annales Geophysicae".

A minor correction. Page 5, line 167: Change "2016" to "2006".

Please also note the supplement to this comment:
https://www.ann-geophys-discuss.net/angeo-2019-55/angeo-2019-55-RC2-supplement.pdf

---

## Author Comment (AC2) · 21 May 2019

We thank the Referee for his appreciation of our work, and also thank him for pointing out the mistake in the paper. We will correct the mistake as soon as possible.

---

## Author Response (AR1)

**possible Stratification phenomenon in ionospheric F2 Layer using the data observed by the D**EMETER** satellite: Method and Results**

Xiuying Wang, Dehe Yang, Dapeng Liu, Wei Chu

Institute of Crustal Dynamics, China Earthquake Administration, Beijing, China

Corresponding author:   Xiuying Wang (652383915@qq.com)

**Response to referees**

**Referees' comments**

**Referee 1**

This paper concerns the study of the electron density from DEMETER satellite (2004- 2010) during the change of altitude (January 2006). The way to establish a possible F2 stratification is quite original, although it cannot provide a definitive response to the question (probably this should be better evidenced in the text). Therefore, the paper deserves publication, with only a very minor corrections, as follows:
   - Please write, in the title and across the whole text, DEMETER with capital letters.
   - Please remove "random" at line 106: the phenomena as geomagnetic storms and sunspot eruptions are not completely random, having some components of quasiperiodicity (27 days, 11 years, etc.).
   - Please correct at line 167: "2016" with "2006"

**Referee 2**

In this work the authors investigate the stratification phenomenon in the ionospheric F2 layer using the nighttime data observed by Demeter satellite in the period from January 1 to 25, 2006. The data were recorded by the Demeter satellite before and after its altitude adjustment that provide to the authors an opportunity to study the vertical gradients of electron density in a small height range (between ï¸A¿677.8 and ï¸A¿669.3 km) of the topside ionosphere using in situ electron density data recorded by the same instrument. It is very important that the period of quiet days (Dst = ±20 nT; F10.7 = 70 ± 5) was chosen by the authors. An original statistical method proposed by authors is used to study the stratification phenomenon. The data observed at different altitudes by the Demeter satellite are compared, and the significance of the differences is checked. As a result, it was found that the electron density data recorded at higher altitude are higher than those of lower altitude, a feature of the stratification phenomenon. Finally, it was found that the stratification occurs in the vicinity of the geomagnetic equator on a global scale

that was interpreted by authors as the stratification of the F2 layer. The paper presents interesting original results and valuable observations which should be published in the "Annales Geophysicae".

A minor correction. Page 5, line 167: Change "2016" to "2006".

**Authors' response**

**Response to Referee 1**

First of all, we would like to thank the reviewer for his affirmation of our work and pointing out the improper expressions in the paper. As the results in this paper is based on all the globally distributed data, the location, local time, and occurrence season etc. of the phenomenon that data at higher altitude are greater than that at lower altitude are consistent with the features of the stratification phenomenon, we therefore deduce it is the stratification phenomenon. Definitive conclusion needs supports from direct observations. For now, there is no direct observations that can obtain the global distribution of the stratification phenomenon. We will collect data from different measurements, and try to validate the result in our follow-up work.

**Response to Referee 2**

We thank the Referee for his appreciation of our work, and also thank him for pointing out the mistake in the paper. We will correct the mistake as soon as possible.

**Modification lists**

According to referee 1 and referee 2:
(1)2016 is modified to 2006.
According to referee 1:
(2)Demeter is modified to DEMETER all over the paper.
(3)Random is deleted from line 106.

[revised manuscript text omitted]

Lebreton, J. P., Stverak, S., Travnicek, P., Maksimovic, M., Klinge, D., Merikallio, S., et al. (2006). The ISL Langmuir probe experiment processing onboard DEMETER: Scientific objectives, description and first results. *Planetary and Space Science*, 54,472-486. https://doi.org/10.1016/j.pss.2005.10.017

Lockwood, G.E.K., & Nelms, G.L. (1964). Topside sounder observations of the equatorial anomaly in the 75W longitude zone. *Journal of Atmospheric and Terrestrial Physics*, 26 (5), 569–580. https://doi.org/10.1016/0021-9169(64)90188-6

Lynn, K.J.W., Harris, T.J., & Sjarifudin, M. (2000). Stratification of the F2 layer observed in Southeast Asia. *Journal of Geophysical Research*, 105(A12):27147-27156. https://doi.org/10.1029/2000JA900056

Nayak, C. K., Yadav, V., Kakad, B., Sripathi, S., Emperumal, K., Pant, T. K., et al. (2014). Peculiar features of ionospheric F3 layer during prolonged solar minimum (2007–2009) . J. Geophys. *Journal of Geophysical Research: Space Physics*, 119, 8685–8696. doi:10.1002/2014JA020135

Raghavarao, R., & Sivaraman, M.R. (1974). Ionization ledges in the equatorial ionosphere. *Nature*, 249, 331–332. DOI:10.1038/249331a0

Rama Rao, P. V. S., Niranjan, K., Prasad, D. S. V. V. D., Brahmanandam, P. S., & Gopikrishna S. (2005). Features of additional stratification in ionospheric F2 layer observed for half a solar cycle over Indian low latitudes. *Journal of Geophysical Research*, 110, A04307. doi:10.1029/2004JA010646

Sayers, J., Rothwell, P., & Wager J. H. (1963). Field Aligned Strata in the Ionization above the Ionspheric F2 Layer. Nature, 198(4877), 230-233. DOI: 10.1038/198230a0

Sen, H.Y. (1949). Stratification of the F2 layer of the ionosphere over Singapore. *Journal of Geophysical Research*, 54(4),363-366. doi:10.1029/jz054i004p00363

Sharma, P., & Raghavarao, R. (1989). Simultaneous occurrence of ionization ledge and counterelectrojet in the equatorial ionosphere: observational evidence and its implications. *Canadian Journal of Physics*, 67(2-3): 166-172. doi:10.1139/p89-028

Skinner, N. J., Brown, R. A., & Wright, R. W. (1954). Multiple stratification of the F-layer at Ibadan. *Journal of Atmospheric and Terrestrial Physics*, 5(1–6), 92–100. https://doi.org/10.1016/0021-9169(54)90013-6

Tardelli, A., Fagundes, P. R., Pezzopane, M., Venkatesh, K., & Pillat, V. G. (2016). Seasonal and solar activity variations of F3 layer and quadruple stratification (StF-4) near the equatorial region. *Journal of Geophysical Research: Space Physics,* 121,12116–12125. doi: 10.1002/ 2016JA023580

Thampi, S. V., Ravindran, S., Devasia, C. V., Pant, T. K., Sreelatha, P., & Sridharan, R. (2005). First observation of topside ionization ledges using radio beacon measurements from low Earth orbiting satellites, Geophysical Research Letters, 32(11). doi:10.1029/2005GL022883

Uemoto, J., Ono, T., Kumamoto, A., & Iizima, M. (2004). Ionization ledge structures observed in the equatorial anomaly region by using PPS system on-board the Ohzora (EXOS-C) satellite. *Earth, Planets and Space*, 56(7), e21-e24. https://doi.org/10.1186/BF03352524

Zain, A. F. M., Abdullah, S., Homam, M. J., Seman, F. C., Abdullah, M., & Ho, Y. H. (2008). Observations of the F3‐layer at equatorial region during 2005. *Journal of Atmospheric and Solar-Terrestrial Physics*,70(6), 918–925. doi:10.1016/j.jastp.2007.12.002

Zhao, B., Wan, W., Reinisch, B., Yue, X., Le, H., Liu, J., & Xiong B. (2011a). Features of the F3 layer in the low‐latitude ionosphere at sunset. *Journal of Geophysics Research*, 116, A01313. doi:10.1029/2010JA016111

Zhao, B., Wan, W., Yue, X., Liu, L., Ren, Z., He, M., & Liu J. (2011b). Global characteristics of occurrence of an additional layer in the ionosphere observed by COSMIC/FORMOSAT-3. *Geophysical Research Letters*, 38(2), L02101. doi:10.1029/2010GL045744